# Specification of the endocrine primordia controlling insect moulting and metamorphosis by the JAK/STAT signalling pathway

**Mar García-Ferrés, Carlos Sánchez-Higueras, Jose Manuel Espinosa-Vázquez, James C-G Hombría** [ID] *

Centro Andaluz de Biología del Desarrollo (CABD), CSIC-JA-UPO, Seville, Spain

* jcashom@upo.es

**Data Availability Statement:** All relevant data are within the paper and its Supporting Information files.

## Abstract

The *corpora allata* and the prothoracic glands control moulting and metamorphosis in insects. These endocrine glands are specified in the maxillary and labial segments at positions homologous to those forming the trachea in more posterior segments. Glands and trachea can be homeotically transformed into each other suggesting that all three evolved from a metamerically repeated organ that diverged to form glands in the head and respiratory organs in the trunk. While much is known about tracheal specification, there is limited information about *corpora allata* and prothorathic gland specification. Here we show that the expression of a key regulator of early gland development, the *snail* gene, is controlled by the *Dfd* and *Scr* Hox genes and by the Hedgehog and Wnt signalling pathways that induce localised transcription of *upd*, the ligand of the JAK/STAT signalling pathway, which lies at the heart of gland specification. Our results show that the same upstream regulators are required for the early gland and tracheal primordia specification, reinforcing the hypothesis that they originated from a segmentally repeated organ present in an ancient arthropod.

## Author summary

The main endocrine organs controlling insect moulting and metamorphosis are the *corpora allata* and the prothoracic glands. Genetic experiments in *Drosophila melanogaster* suggested that, despite their extremely different morphology and function, the *corpora allata* and the prothoracic glands are homologous to the respiratory trachea. All three organs derive from a primordium arising at similar locations along the cephalic and trunk segments, they activate common developmental genes using the same *cis*-regulatory elements, and can be transformed into each other by modifying Hox expression. One key difference between glands and trachea is that the endocrine primordia activate the Epithelial to Mesenchymal inducer gene *snail*. Using the *snail* gland specific enhancer as a proxy for gland formation, we show that the glands are specified by the same inputs specifying the trachea. These include the JAK/STAT, the Hedgehog and Wingless signalling

**Funding:** This work was supported by the Spanish Ministerio de Ciencia e Innovación (MICINN) and European Regional Development Fund (FEDER) grant PID2019-104656GB-I00 to MGF, CSH, JMEV and JCGH; grant BES-2017-081120 to MGF; the Consejería de Economía, Innovación, Ciencia y Empleo, Junta de Andalucía (Department of Economy, Innovation, Science and Employment, Government of Andalucia) grant P20-0003 to CSH and JCGH and the María de Maeztu Unit Excellence Grant CEX-2020-001088-M to MGF, CSH, JMEV and JCGH. The funders had no role in study design, data collection and analysis, decision to publish, or preparation of the manuscript.

**Competing interests:** The authors have declared that no competing interests exist.

pathways as well as inputs from the Hox genes. These observations support the hypothesis that during arthropod evolution, a metamerically repeated organ diverged to give rise to endocrine glands in the head and respiratory organs in the trunk segments.

## Introduction

Arthropods are characterised by the presence of an external skeleton that protects them from injury but also constrains their growth during development. This problem is solved by a dedicated endocrine system controlling the periodic moulting of the exoskeleton. Two glands control the process of larval moulting and metamorphosis in insects: the *corpora allata* (CA), which secrete Juvenile Hormone; and the prothoracic glands (PG), which secrete Ecdysone [1]. In holometabolous insects, secretion of both of these hormones to the haemolymph induces the larva moulting into a larger larva, while secretion of Ecdysone alone induces metamorphosis [2]. Similar endocrine glands secreting hormones related to those produced by the CA and the PG have been identified in crustaceans, indicating that this system has an ancient evolutionary origin [1,3,4].

Analysis of development in *Drosophila melanogaster* showed that the CA and the PG primordia are specified in the lateral ectodermal cells of the maxillary and the labial segment respectively, at homologous locations to those giving rise in more posterior segments to the fly's respiratory organs [5]. During early development, the CA and the PG primordia exhibit a similar behaviour to that of the tracheal primordia, with the epithelium invaginating to form small sacks of cells resembling tracheal pits. However, while the tracheal primordia maintain an epithelial organization throughout development, the gland cells soon experience an Epithelial to Mesenchymal Transition (EMT) induced by *snail* (*sna*) gene expression [5]. Following Snail activation, the CA and the PG coalesce into a single primordium that migrates across four segments until it reaches the dorsal part of the first abdominal segment (A1). This migration is guided by several intermediate landmarks that serve as "stepping stones" during their long-range migration (Fig 1A) [6]. Once in A1, the CA/PG primordium fuses ventrally to the *corpora cardiaca*, an independent endocrine organ of mesodermal origin [7,8], and dorsally to the contralateral primordium, giving rise to a ring structure encircling the anterior aorta. Therefore, the mature ring gland is a composite endocrine organ formed by three different glands, two of ectodermal origin (the CA and the PG) and one of mesodermal origin, the *corpora cardiaca* [1,6].

Despite their different morphology and function, the CA and the PG have several characteristics in common with the trachea. First, the CA and the PG are specified in the cephalic lateral ectoderm at homologous positions to those forming the tracheal primordia in the trunk segments. Second, all three organs express the gene encoding the transcription factor Ventral veinless (Vvl) activated through the same enhancer (*vvl1+2*). Third, ectopic expression of the *Deformed* (*Dfd*) or the *Sex combs reduced* (*Scr*) Hox genes can transform tracheal primordia cells into gland cells and, conversely, the ectopic activation of trunk Hox genes can transform the gland primordia into trachea. These observations led to the proposal that the CA, the PG and the trachea arose from a metamerically repeated ancient structure that evolved divergently in each segment giving rise to three completely different organs [5]. This hypothesis has been reinforced by functional studies performed in the *Oncopeltus* hemipteran insect [9].

In comparison to the extensive knowledge we have of the mechanisms specifying the *Drosophila* tracheae [10–20], little is known about CA and PG specification. The first signs of CA and PG specification are noticeable when these primordia start expressing the *sna* gene [5].

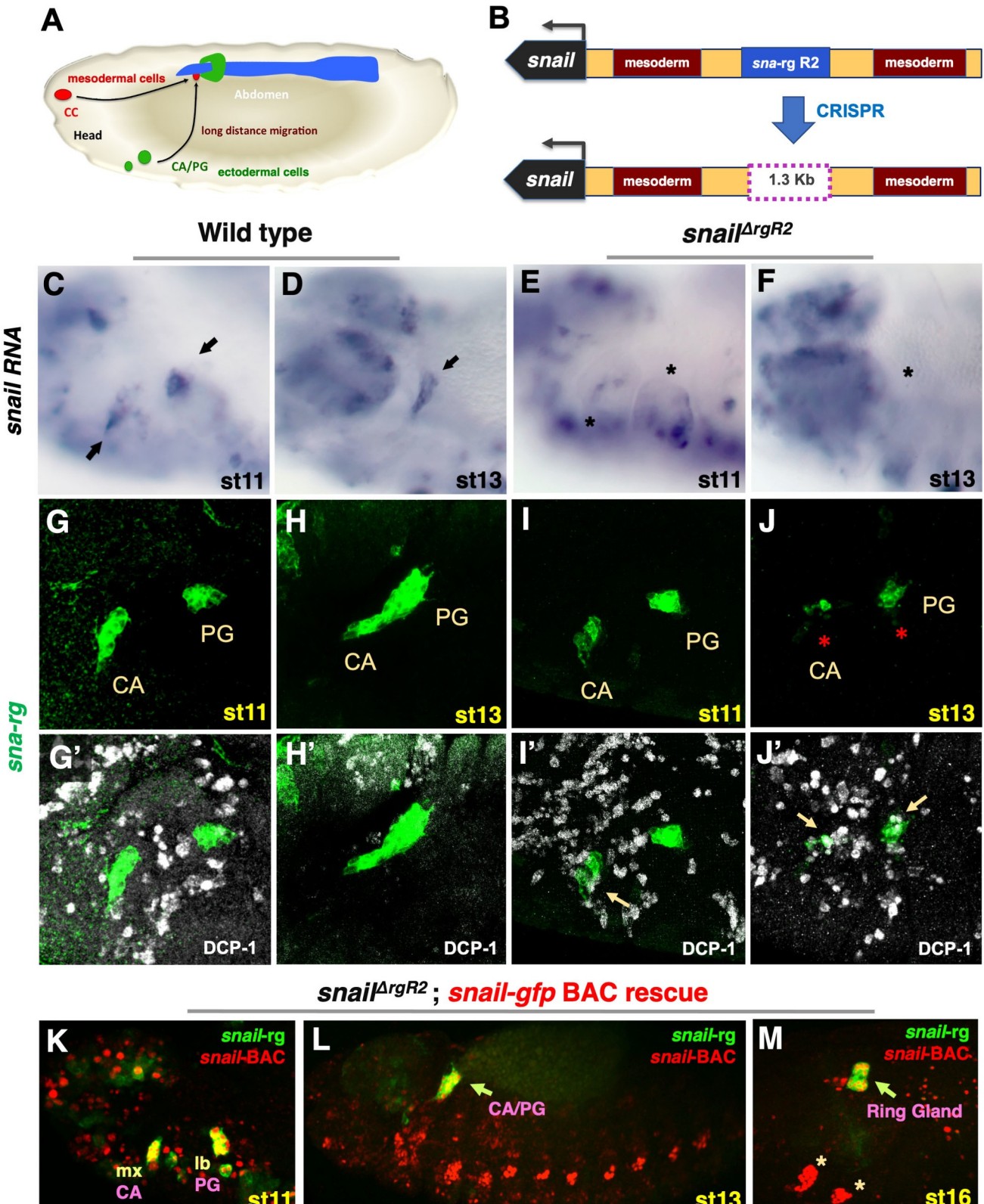

**Fig 1. Localisation of the specific *corpora allata* and prothoracic *snail cis* regulatory elements. (A)** Scheme of a st16 embryo representing the CA and PG in green, the *corpora cardiaca* (CC) in red and the aorta and heart in blue. The migratory route followed by the three gland primordia towards their

final position in the ring gland is represented by arrows starting from their approximate location at st11. **(B)** *sna* locus indicating the position of the transcription unit (black), the two mesoderm enhancers (brown), the ring gland enhancer (blue) and the *sna^{ΔrgR2}* deletion. **(C-F)** *sna* RNA expression in wild type embryos at st11 (C) and st13 (D), or *sna^{ΔrgR2}* embryos at st11 (E) and st13 (F). Arrows point to the CA and PG primordia, asterisks mark the absence of *sna* transcription. **(G-H)** *sna-rg-GFP* reporter in a st11 wild type embryo (G) before CA and PG coalescence, and at st13 (H) showing the coalesced CA/PG migrating towards the dorsal midline. **(I-J)** *sna-rg-GFP sna^{ΔrgR2}* homozygous embryos showing the CA and PG primordia at st11 (I) and at st13 (J) when degeneration is noticeable (asterisks). **(G'-J')** Show DCP-1 co-expression (white) in the same embryos to reveal apoptosis. In control embryos (G'-H') DCP-1 activation is restricted to ectodermal cells. In *sna-rg-GFP sna^{ΔrgR2}* homozygous embryos, gland cells show high levels of DCP-1 at st13 (J') (yellow arrows). At st11 (I'), just after gland specification, DCP-1 starts being detectable before overt gland degeneration. **(K-M)** *sna^{ΔrgR2}* homozygous embryos carrying the *sna-rg-mCherry* reporter (green) and a *sna-BAC* rescue construct. The Sna protein in the BAC is tagged with GFP (red) revealing its expression in the gland primordia before coalescence (K, yellow because overlap with *sna-rg* expression), after coalescence (L, arrow), and after integrating in the ring gland (M). Apart from the gland primordia, the Snail BAC protein reveals other sites of expression: the oenocytes and the wing and haltere primordia (M, asterisks). Note that cell viability and migratory behaviour of the CA and PG are fully rescued by the BAC. All figures show lateral views with anterior left and dorsal up.

Snail is a zinc-finger transcription factor conserved in vertebrates where its function has also been associated to the induction of EMT [21–23]. Apart from its function in the endocrine primordia, Snail is also required for the formation of the mesoderm [24,25]. The *sna-rg-GFP* reporter gene, made with a 1.9 kb *sna cis*-regulatory element, is the earliest known specific marker for the CA and the PG primordia [5]. *sna-rg-GFP* expression is first activated at the beginning of organogenesis (st11), after the two gland primordia have just invaginated in the maxillary and labial segments, and its expression is maintained throughout embryonic gland development (Fig 1G and 1H). Thus, *sna* expression is a CA and PG specific marker comparable to what *trh* expression is for the trachea. Both genes encode transcription factors labelling the respective primordia at the earliest stages of development and both genes are required for the development of the organs where they are activated. Therefore, finding the upstream regulators of *sna-rg* expression should help uncovering the mechanisms required for gland specification. Moreover, the comparison of the gene network activating *sna* expression in the gland with that activating *trh* expression in the trachea will allow us to confirm if both organs share similar upstream regulators as would be expected if they shared a common evolutionary origin.

To find out what are the mechanisms inducing CA and PG specification we have analysed how *snail* expression is activated in the primordia of these organs. We show that the Wnt and Hh pathways determine the antero-posterior segmental location where the *sna-rg* enhancer is activated. This is achieved indirectly through the localised transcriptional activation of the *upd* gene, which encodes a ligand activating the JAK/STAT signalling pathway. We show STAT directly activates *sna* expression in the glands and propose that the Hox input required for activating *sna* expression is mediated indirectly.

## Results

### *sna* expression in the CA and PG primordia is activated by a single *cis*-regulatory region

Expression of the *snail* gene in the *corpora allata* (CA) and the prothoracic gland (PG) primordia is key for their specification and development [5]. To test if the *sna-rg cis*-regulatory region previously described is the only element activating *snail* expression in the CA and the PG primordia, we created *sna^{ΔrgR2}*, a deletion generated with the CRISPR-Cas9 system using specific single guide RNAs (Figs 1B and S1 and Materials and Methods). RNA *in situ* hybridization reveals *sna^{ΔrgR2}* embryos lack *sna* expression in the CA and PG primordia while maintaining it in other organs (Fig 1C–1F).

Embryos homozygous for *sna^{ΔrgR2}* or heterozygous for this deletion over the *sna^1* null allele are not viable. These embryos develop a normal mesoderm with the only obvious phenotypic

defect being the almost complete degeneration of the CA and the PG primordia (Fig 1G–1J). Embryo lethality and gland development are fully restored by a *sna-GFP BAC* construct [26], revealing that *sna^ArgR2* lethality is due to the *sna* deletion (Fig 1K–1M).

These results prove that the *sna^ArgR2* deletion inactivates the only regulatory region driving *sna* expression in the CA and PG gland primordia, allowing us to use *sna-rg-GFP* reporter expression as a proxy to discover the upstream trans regulatory elements involved in *sna* transcription and CA and PG specification.

## Requirement of the Wnt signalling pathway for gland specification

The *vvl* and *sna* genes are co-expressed in the CA and the PG, but the expression of *sna* in the gland primordia does not depend on Vvl function [5], suggesting that both genes may respond to similar upstream regulatory cues in the gland region. As tracheal *vvl* expression expands in *wingless* (*wg*) mutants [20], we tested if *sna-rg* spatial activation is also restricted through the Wnt signalling pathway. In *wg^CX4* or in *wg^en11* homozygous mutant embryos *sna-rg-GFP* expression in the maxilla and the labium appears duplicated at st11 (Fig 2A and 2B). The duplicated primordia form in cells normally expressing Wg and are located at the same dorso-ventral position where the endogenous primordium of that segment forms. The ectopic and the normal *sna-rg* expressing cells become migratory coalescing into a single larger gland primordium, suggesting the ectopic cells form functional gland primordia, although this expanded primordium cannot reach the embryo's dorsal side due to the general defects in *wg* mutants.

Ectopic *UAS-wg* expression driven in the maxilla and labium with the *sal-Gal4* driver eliminates *sna-rg* reporter expression (Fig 2C). This repression is mediated through the Wnt canonical pathway as *sna-rg-GFP* expression is also eliminated by ectopic expression of an activated form of Armadillo (*UAS-ArmS10*, Fig 2D) [27]. Surprisingly, we found that while *sna-rg* expression is normal in embryos homozygous for the *pan^2* zygotic null allele of dTCF [a.k.a. Pangolin [28,29]], the DNA binding protein downstream of the Wg signalling pathway (Fig 2E), double mutant *wg^CX4*, *pan^2* embryos lack the ectopic gland primordia but not the endogenous ones (Fig 2F). These results suggest that Arm-dTCF can prevent *sna-rg* expression in Wg expressing cells but it does not affect the formation of the endogenous gland primordia which are out of Wg signalling range.

## Requirement of the Hedgehog (Hh) signalling pathway for gland specification

It has been reported that *vvl* expression in the tracheal primordia is strongly reduced in *hh* mutants [12]. The Hh and Wnt signalling pathways cross-regulate in the trunk epidermal cells where Hedgehog signalling is required for maintenance of *wg* expression in the adjacent ecto-dermal cells of the anterior compartment, and Wg signalling is required for the maintenance of *hh* and *engrailed (en)* expression in the posterior compartment [30]. As a result of this cross-regulation, *wg*, *en* and *hh* mutant embryos have similar phenotypes in the trunk ventral ectodermal segments [31]. However, in the cephalic region, where the glands are specified, such cross regulation does not occur, with Engrailed expression being maintained in the posterior segments of the maxilla and the labium in the absence of *wg* function [32]. To study the effect of Hh signalling on gland development, we analysed *hh^AC* and *en^E* homozygous mutant embryos and found an almost complete absence of *sna-rg* expression (Fig 3B and 3C). Engrailed activates *hh* expression in the posterior compartment, from where secreted Hh induces the pathway in neighbouring cells. The final target is the Cubitus interruptus (Ci) protein that can act either as a transcriptional activator or as a repressor depending on the

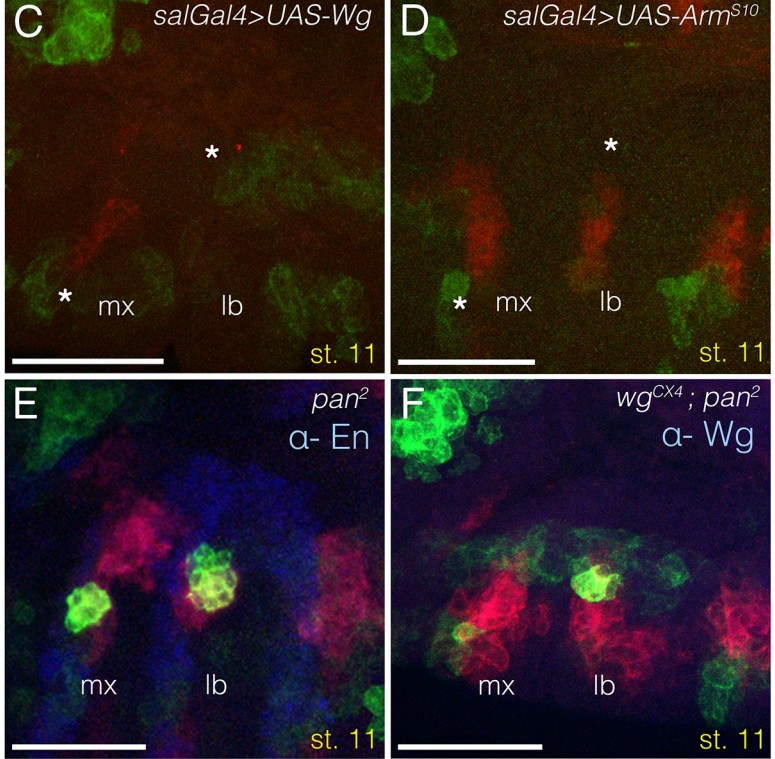

**Fig 2. Wnt pathway requirement for gland specification.** (A-F) st11 gland primordia double labelled with *sna-rg-GFP* in green, and in red the *wg* expressing cells with a *wg*[en11] reporter line (A-B) or the *vvl* expressing cells with a *vvl1 +2-mCherry* reporter line (C-F). **(A)** Control *sna-rg-GFP wg*[en11]*-lacZ/+* embryo. *sna-rg* labels specifically the CA and the PG cells invaginating at the anterior end of the maxillary and labial segments respectively. **(B)** In *wg*[en11] homozygous mutants, ectopic patches of *sna-rg* expression appear on the *wingless* expressing cells (arrows) in the maxilla and labium at the same dorso-ventral positions as the endogenous ones. **(C)** Wg ectopic expression driven in the maxillary and labial segments with *spalt-Gal4* results in the downregulation of *sna-rg* expression (asterisks). Downregulation of *vvl1+2* cephalic expression is also observed. **(D)** Ectopic expression of activated Armadillo results in *sna-rg* downregulation (asterisks). **(E)** *pan²(dTCF)* null mutants present normal *sna-rg* expression and Engrailed (blue) stripe expression is maintained. **(F)** *wg*[CX4] *pan²* double mutant embryos do not present ectopic endocrine primordia. Embryo in panel F is also stained with anti-Wg to recognise the homozygous *wg* mutants. Scale bar 50 μm.

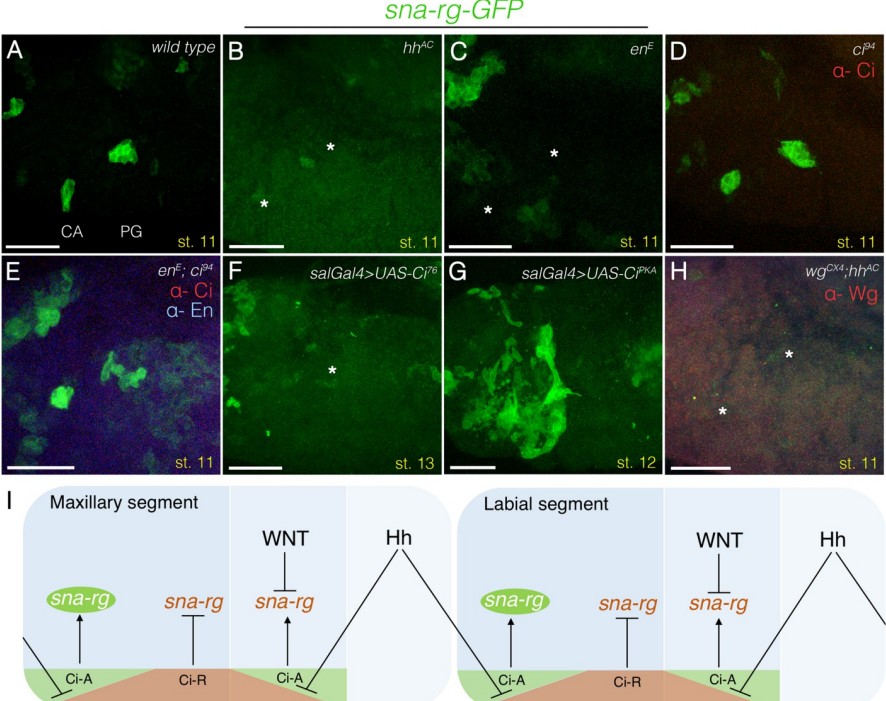

**Fig 3. Hh pathway requirement for gland specification.** (A-H) Embryos of various genotypes labelled with *sna-rg-GFP* (green) to mark the gland primordia. **(A)** Wild type embryo labelling the CA and PG location. **(B)** *hh^AC* null and **(C)** *en^E* null embryos show almost complete downregulation of the *sna-rg* enhancer (asterisks). **(D)** *ci^94* null embryos show normal *sna-rg* activation. **(E)** Double *en^E*, *ci^94* mutants recover *sna-rg* expression in the CA and PG primordia. **(F)** Overexpression of the Ci repressor isoform with *spalt-Gal4* downregulates *sna-rg* expression (asterisk marks the approximate position where stage 13 migrating primordia should be expected). **(G)** Overexpression of the Ci activator isoform with *sal-Gal4* results in an expansion of *sna-rg* expression. **(H)** Double *wg^CX4*, *hh^AC* mutant embryos lack *sna-rg* expression (asterisks). **(I)** Model summarising regulatory interactions between the Wnt and Hh pathways and the *sna-rg* enhancer (The interactions represented are not assumed to be direct). Panels D-E were also stained with anti-Ci, panel E with anti-En and panel H with anti-Wg to identify the mutant embryos. All embryos are at st11 except F and G which are at st13 and st12. Scale bar 50 μm.

pathway's activation state. In the absence of Hh, Ci is cleaved giving rise to a protein repressing the transcription of its direct targets [33]. Conversely, in the presence of Hh, the pathway's activation prevents Ci's cleavage, giving rise to a transcriptional activator [34].

We find that in *ci^94* null embryos *sna-rg-GFP* is expressed in its normal pattern (Fig 3D), indicating Ci is not a necessary activator of *sna* expression in the glands. We also found that in double *en^E*, *ci^94* mutant embryos the *sna-rg-GFP* expression is recovered compared to *en^E* embryos (Fig 3 compare panel C with E), indicating that the Ci repressor form prevents *sna-rg* activation. To confirm this, we expressed *UAS-Ci76*, the repressor isoform of Ci [33], with the s*al-Gal4* line and found this causes an almost complete absence of *sna-rg* activity (Fig 3F). Although the above results indicate Ci is not absolutely required for *sna-rg* expression, we observed that overexpression of Ci^PKA, the active form of Ci, causes a non-fully penetrant expansion of *sna-rg* expression (Fig 3G) suggesting the possibility that *sna-rg* may be responsive to Ci and to a second activator. We also analysed double *wg*, *hh* (or *wg*, *en*) mutants and found that these embryos do not activate *sna-rg*, a phenotype similar to that of *hh* mutants (Fig 3H). These results indicate that Ci repression is epistatic over the derepression caused in *wg* mutants.

The above data fit a model where *sna-rg* expression is under negative regulation, either directly or indirectly, mediated by the Wnt and Hh signalling pathways (Fig 3I). Although Ci repression of *sna-rg* activity should be relieved by Hh signalling anteriorly and posteriorly to the En expressing cells, the Wnt parallel repressive function prevents *sna-rg* activation in Wg expressing cells restricting the formation of the CA and PG primordia to the most anterior cells of the maxillary and labial segments.

## Regulation of Upd ligand expression by the Wg and Hh pathways

Previously we showed that JAK/STAT signalling is required for *sna-rg* expression [5]. To find out if the Wg and Hh signalling pathways regulate *sna* indirectly via JAK/STAT signalling, we reanalysed the spatio-temporal activation of *upd* in wild type and mutant embryos, paying special attention to the maxillary and labial segments where the gland primordia are specified. In st9 wild type embryos, *upd* is expressed in segmental stripes immediately posterior to the Engrailed expressing cells (S2 Fig). This pattern of transcription evolves to form a transient antero-posterior lateral stripe that rapidly resolves at early stage 11 into two patches of expression in the maxilla and labium corresponding to the sites where the CA and PG glands form (Figs 4C and S2E). Expression analysis of *10xSTAT-GFP*, a reporter that is universally activated in cells where the JAK/STAT pathway is active [35,36] confirms JAK/STAT signalling activation at st10 and 11 in the CA and PG primordia (Fig 4A–4B). Although *upd* is transcribed in both primordia, we noticed that expression of both *upd* RNA and the *10XSTAT-GFP* reporter is more transient in the CA than in the PG primordium (Fig 4A–4D).

We next analysed if the Wnt and the Hh pathways affect *upd* transcription in the gland primordia. In *hh*[AC] null embryos, we find that the transient *upd* expression in the CA and PG primordia disappears (Fig 4E), while in *wg*[CX4] mutants *upd* RNA expression expands (Fig 4F). We also found that ectopic expression of the activator Ci protein results in a non-fully penetrant expansion of *upd* expression in stage 10 embryos (Fig 4H–4I). These results suggest that the effects on *sna-rg* expression caused by mutations affecting the Wnt and Hh signalling pathways are mediated indirectly through the JAK/STAT signalling pathway.

## Possible cross-regulation between *Hox*, *wg*, *hh* and *upd* in the maxillary and labial segments

Development of the CA and PG and normal expression of the *sna-rg* reporter in the maxilla and the labium require Dfd and Scr function [5], therefore we studied if there are any cross-regulatory interactions among the genes involved in gland primordia specification.

We first analysed *wg* and *en* mutant embryos and found that the expression of Dfd and Scr is not significantly affected (S3A–S3F Fig). Similarly, neither En nor Wg expression is affected in *Dfd Scr* mutant embryos (S3G–S3J Fig), discarding a possible interaction between Dfd and Scr and the Wnt/Hh signalling pathways. In contrast, we found that the transient expression of *upd* transcription in the CA and PG primordia almost disappears in *Dfd Scr* mutant embryos (Fig 4G), indicating that the Hox proteins can regulate JAK/STAT signalling as previously shown for Abd-B [37]. These results indicate that the Hox, and the Wnt/Hh pathways indirectly mediate the regulation of the *sna-rg* enhancer through their modulation of *upd* expression and JAK/STAT signalling activation.

The *sna-rg* reporter is not expressed in *Df(1)os1A* embryos (Fig 5A and 5B). To test if generalised Upd expression in the maxilla and labium can activate *sna-rg* independently of other upstream positive or negative inputs, we induced *UAS-upd* with either the *sal-Gal4* or the *arm-Gal4* lines. We observe that these embryos have expanded *sna-r*g expression along the antero-posterior axis in the maxillary and labial segments (Fig 5C). Analysis of Sal expression, which

**Fig 4. JAK/STAT pathway requirement for gland specification. (A-B)** Wild type embryos carrying the *sna-rg-mCherry* (red) and *10xSTAT-GFP* (green) reporter genes. (A) At st10, embryos present patches of green staining in the maxillary and labial segments at the position where the gland primordia will become activated. Weak *sna-rg* expression starts to appear. (B) At st11, when *sna-rg-mCherry* is strongly activated, *10xSTAT-GFP* expression has faded from the maxilla while is maintained in the labium. Note that *10xSTAT-GFP* expression is also activated in the tracheal primordia. (C-I) *upd* mRNA *in situ* hybridization in st10-st11 embryos. (C) A wild type early st11 embryo showing *upd* mRNA in the maxillary and labial segments is restricted to two patches where the CA and the PG form. **(D)** At late st11, *upd* mRNA expression only remains in the PG (asterisk marks the position where the CA primordium is located). **(E)** *hh* null mutants lose maxillary and labial *upd* transcription (asterisks), which remains in the tracheal primordia. **(F)** In *wg* null mutants, *upd* transcription is extended in the maxillary and labial segments (arrows), as well as expanding

around the tracheal primordia. **(G)** *Dfd Scr* mutants lose maxillary and labial *upd* transcription (asterisks). Embryos in (G) are also *Abd-B* homozygous mutant to allow distinguishing unambiguously the triple *Dfd Scr Abd-B* homozygous embryos by the absence of *upd* expression in the A8 posterior spiracle primordium. (H) *upd* expression in a control embryo at st10. (I) *upd* expression in an embryo expressing the activator isoform of Ci in a st10 embryo. Scale bars 50 μm.

labels the PG primordium [5] shows that Upd ectopic expression induces a moderate expansion of the CA primordia while resulting a much larger increase of the PG primordium (Fig 5D and 5E). This expansion occurs mostly in the antero-posterior axis from cells where the Hh and the Wnt pathways are normally blocking *sna-rg* expression, while expansion is less noticeable in the dorso-ventral axis. This indicates that most of the antero-posterior intrasegmental inputs provided by the segment polarity genes converge on Upd transcription but that the dorso-ventral information is registered downstream of Upd.

We finally tested if activation of *UAS-upd* with the *sal-Gal4* driver line can rescue *sna-rg* activation in *Dfd Scr* mutant embryos. We found that the residual levels of GFP observed in *sna-rg Dfd Scr* mutant embryos are not increased in *sal-Gal4 UAS-upd sna-rg Dfd Scr* embryos (Fig 5F and 5G), indicating that besides regulating *upd* expression, the Hox input has further requirements for gland formation.

Therefore, localised Upd expression defines the antero-posterior intrasegmental localisation of the CA and PG primordia, but other signals besides STAT must be controlling the dorso-ventral and the cephalic *sna* activation either directly at the *sna-rg* enhancer level or through unknown intermediate regulators.

## Analysis of the direct regulation of *sna-rg* enhancer by STAT

To find out if the Hox and STAT inputs regulate *sna* expression directly, we searched for putative binding sites in the *cis*-regulatory region of *sna-rg*. To facilitate the bioinformatic analysis we dissected the 1.9 kb *sna-rg* regulatory element down to a 681bp fragment we call R2P2 (S1A–S1E Fig). The *sna-rg-R2P2-GFP* reporter construct drives high levels of expression in the CA and PG and its expression is even more specific as it lacks the low levels of GFP expression observed in the haemocytes and neurons of the larger *sna-rg-GFP* reporter.

Computational JASPAR analysis [38] of the 681bp *R2P2* sequence identified three putative Hox-Exd-Hth and three putative STAT binding sites (Fig 6A). Further subdivision of *sna-rg-R2P2* in two halves shows that neither the A1 nor the A2 half drive embryonic expression (Fig 6C and 6D). Reporters containing A1 fused to either the proximal part of A2 (the *sna-rg A1+A2prox-GFP* reporter containing a single STAT site) or to the distal part of the A2 element (the *sna-rg A1+A2dist-GFP* reporter containing two STAT sites) recovered ring gland expression (Fig 6E and 6F). The recovery of expression when A1 is fused to either fragment, both containing STAT binding sites, made us wonder if the lack of expression of the A1 fragment is due to the absence of STAT binding sites. To test this hypothesis, we added to A1 a 20bp fragment that contains a single functional STAT site taken from an unrelated gene [the *vvl1+2* enhancer of the *ventral veinless* gene [20]] creating the *sna-rg A1+STAT* reporter. We find that *A1+STAT* drives expression in both the maxilla and the labium and that this depends on JAK/STAT signalling, as mutation of the STAT-binding site abolishes expression in the *sna-rg A1+STATmut* reporter (Fig 6G and 6H). Taken together, these experiments show that the presence of functional STAT binding sites is required for *sna* activation in the CA and PG primordia and that the 300bp A1 fragment can interpret the segmental cephalic positional information, suggesting that the Hox-Exd-Hth site located in A1 could mediate Dfd and Scr input to the enhancer.

**Fig 5. Epistatic relationship between JAK/STAT, Wnt, Hh and Hox inputs over sna-rg regulation.** (A-C) Embryos expressing *sna-rg-GFP* (green) and *vvl1+2-mCherry* (red) stained with anti-En (blue). **(A)** *sna-rg* and *vvl1+2* expression in st11 control

embryos. Panels (A') and (A'') show each channel separately to appreciate the co-expression of both markers in the gland primordia. **(B)** *Df(1)os1A* embryos show an almost complete downregulation of *sna-rg* and *vvl1+2* expression from the CA and PG (asterisks). **(C)** Ectopic Upd expression driven with *sal-Gal4* induces ectopic *sna-rg* and *vvl1+2* expression in the gnathal segments, which for *sna-rg* is more pronounced in the labium than in the maxilla. Note that in the maxillary segment Upd can induce ectopic dorsal *vvl1+2* but not *sna-rg* expression, this is expected as Dfd only induces *sna-rg* ventrally in the maxilla. (D-E) *sna-rg-GFP* embryos stained with anti-GFP (green) and anti-Sal (red). In control embryos **(D)** Sal labels the PG primordium but not the CA. In *arm-Gal4* embryos ectopically expressing Upd **(E)**, the PG is more expanded than the CA as shown by the number of cells co-expressing Sal and GFP. (F-G) *sna-rg-GFP* expression (green) in st13 *Dfd Scr* mutant embryos **(F),** or *Dfd Scr* mutant embryos after ectopic Upd expression driven with the *sal-Gal4* line **(G)** showing that Upd activation is not sufficient to rescue gland formation in *Dfd Scr* mutants. In *Dfd Scr* mutant embryos **(F)**, although the gland primordia become apoptotic, residual GFP expression indicates that there must exist Hox independent inputs activating the *sna-rg* enhancer. Embryos in (F-G) are also stained with anti-Scr to recognise the homozygous mutants. Scale bars 50 µm.

To confirm STAT's binding sites requirement, we mutated all three putative sites on the *R2P2* fragment generating the *sna-rg-R2P2 STATmut* construct expressing simultaneously the LifeActin-GFP and nuclear Histone-RFP reporter markers (Materials and Methods). Comparing its expression to the *sna-rg-R2P2-eGFP-PH*, we find that although mutating the three STAT binding sites completely abolishes the reporter's expression in the PG, surprisingly, it does not eliminate its expression from the CA, where its activation is only slightly delayed (Fig 6I and 6J). The CA expression of the *sna-rg-R2P2 STATmut* construct still depends on *upd* activity as it disappears in *Df(1)os1A* embryos lacking all Upd ligands (Fig 6K). These results indicate that either there is a cryptic STAT site in *sna-rg-R2P2* we did not mutate, or that in the CA the *sna-rg-R2P2* enhancer can be activated both directly and indirectly by STAT through a site present in the A2 fragment (see discussion).

## Analysis of the regulation of *sna-rg* enhancer by Hox proteins

To test genetically the requirement of the Hox proteins and their cofactors for *sna* activation, we studied *sna-rg* expression in mutants for *Dfd Scr* and for the Hox cofactor *hth* [39]. In *Dfd Scr* mutant embryos few cells activate *sna-rg-GFP* expression at st11 (S4A Fig) and those that do soon acquire an apoptotic aspect (Fig 5F), confirming Hox requirement for gland development. Similarly, in *hth^P2* mutants, *sna-rg* expression almost disappears (S4B Fig). As described above, there are three JASPAR predicted Hox-Exd-Hth putative binding sites present in the *R2P2* fragment. We first mutated the sites located in *sna-R2P2* closer to the STAT binding sites in the A2 region and found that the expression of the mutated construct was almost identical to the wild type fragment. The dispensability of these two Hox-Exd-Hth sites for *sna* activation in the maxilla and labium is further confirmed by the strong expression driven by the *snaA1+A2prox-GFP* reporter that lacks these two sites (Fig 6E). Although the *snaA1+A2prox* reporter construct is slightly derepressed in the cephalic region, it is still active in the CA and PG primordia, indicating that, if there is any direct requirement for Hox activation, this would be mediated by the Hox-Exd-Hth site located in fragment A1. This site has a class 2 sequence (TGACAAAT) that has been shown by SELEX-seq analysis to bind preferentially Dfd and Scr proteins in complex with the Exd-Hth cofactors [[40] and S4C Fig]. We mutated in *snaA1+A2prox* this class 2 site to (TGA**TC**AAT) which is not detected *in vitro* by any Hox-Exd protein complex and found the embryos maintain robust expression in the CA and the PG suggesting the enhancer is not a direct Hox target (S4D' Fig). To confirm this, we also mutated the putative class 2 site in *snaA1+A2prox* changing its affinity to class 1 or class 3 Hox proteins. Such changes have been shown to affect the spatial expression of *vvl1+55*, a reporter construct directly regulated by the Dfd and Scr proteins and as result only active in the maxilla and the labium. While mutating in *vvl1+55* the class 2 site towards a class 3 site conferring affinity for either Antp, Ubx, Abd-A and Abd-B protein in complex with Exd strongly activates

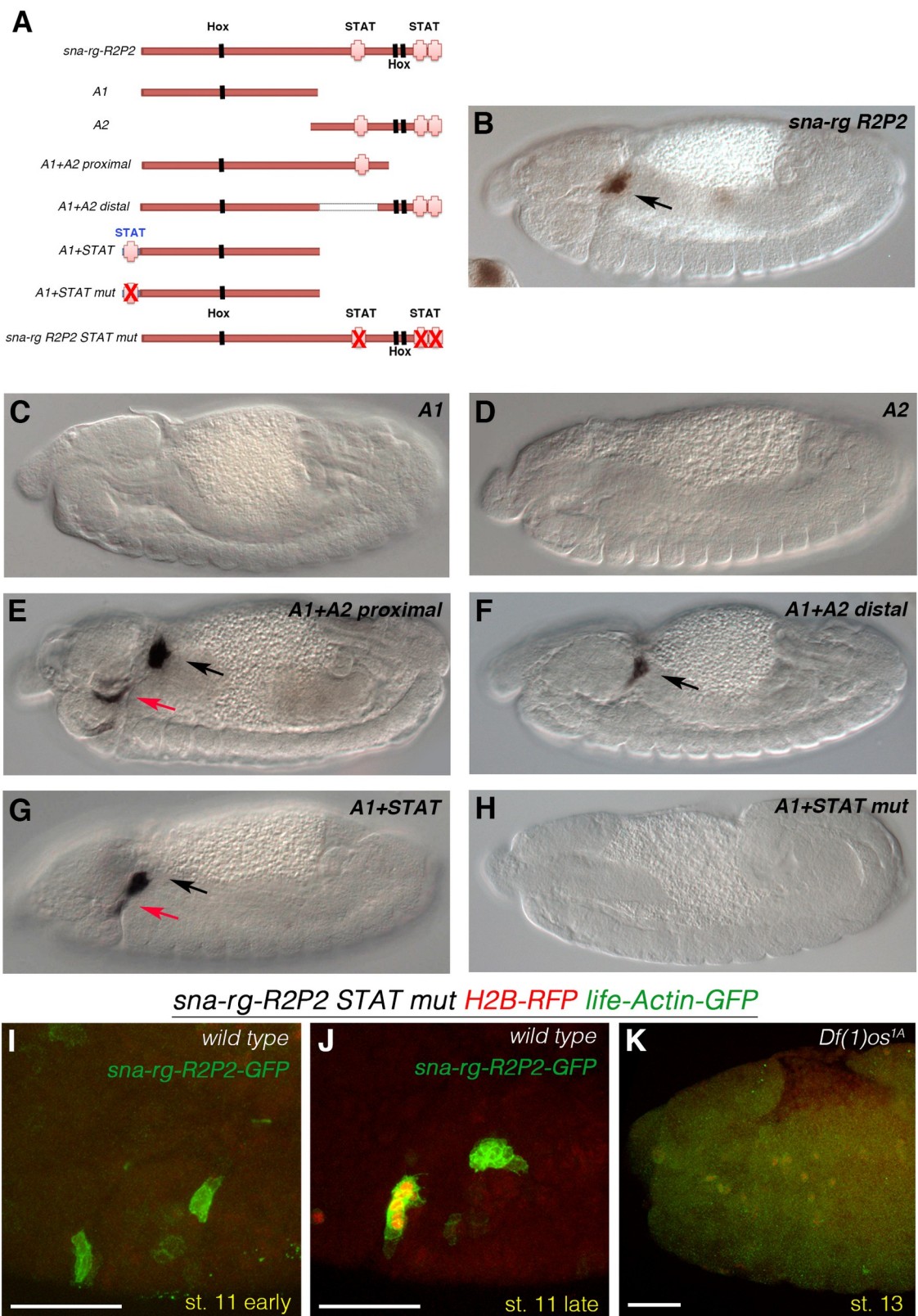

**Fig 6. Direct regulation of *sna-rg* by STAT. (A)** Representation of the minimal *sna-rg R2P2* subfragments indicating the location of the putative STAT (pink crosses) and Hox-Exd-Hth DNA binding sites (black boxes). Mutated STAT binding sites are represented

with a red X over the pink cross. (**B**) *sna-rg R2P2-GFP* expression. (**C**) No GFP expression is observed in A1 nor in A2 constructs (**D**). Gland expression is observed when A1 is joined to the A2 proximal half (**E**) or when A1 is joined to the A2 distal half (**F**). (**G**) A1 fused to a 20bp from fragment the *vvl1+2* enhancer containing a functional STAT binding site. (**H**) A1 fused to the same 20bp fragment where the STAT binding site has been mutated. (I-K) Embryos carrying both the *sna-rg-R2P2-GFP-PH* (green) and the *sna-rg-R2P2-STATmut Histone2B-RFP-GFP-PH* (red and green) constructs. Red nuclear expression is not observed at early stage 11 (**I**) but can be detected later exclusively in the CA (**J**). *Df(1)os1A* embryos lacking all Upd ligands (**K**) do not express *sna-rg-R2P2-STATmut mCherry-GFP-PH*. Black arrows in (E-G) point to the CA/PG gland primordia, red arrows to ectopic expression outside the glands. Scale bars 50 μm.

the enhancer in the trunk, and mutating the sequence towards a class 1 site that confers affinity for the Lab protein activates the enhancer in the intercalary segment where Labial is expressed [19]; equivalent mutations of the class 2 site in the *A1+A2prox* fragment did not modify significantly the spatial expression of the reporter, which remains expressed mostly in the maxilla and labium (S4E–S4G Fig), further supporting that *sna* is not directly activated by the Hox-Exd-Hth complex in the endocrine glands.

## Discussion

### Intrasegmental specification of the CA and the PG

We have found that the Wnt, the Hh and the JAK/STAT signalling pathways contribute to the specification of the CA and the PG in the maxillary and labial segments. Our results indicate that the Hh and the Wnt pathways act indirectly by negatively regulating the spatial activation of the Upd ligand (Fig 7). Engrailed activation of *hh* transcription in the posterior compartment of the cephalic segments leads to Hh diffusion to the neighbouring cells in the maxilla and labium. Hh pathway activation prevents the formation of the Ci repressor protein allowing the activation of *upd* transcription at both sides of the posterior compartment. However, anterior to the *engrailed* stripe, Wnt pathway activation prevents *upd* transcription. As a result of the combined Hh and Wnt inputs, *upd* can be briefly transcribed at stage 11 in two localised ectodermal patches from where it induces JAK/STAT signalling, which activates *sna* transcription in the CA and the PG primordia through the *sna-rg* enhancer.

*sna* transcription in the endocrine primordia is mediated through a 681bp non-redundant *cis*-regulatory element located 5.439bp upstream of the *sna* transcription unit. The *sna-rg-R2P2* enhancer contains a STAT 3n and two STAT 4n binding sites conforming to the canonical sequence TTCNNN(N)GAA [41,42]. Deletion from the *R2P2* enhancer of a 278bp fragment containing all three putative STAT binding sites results in a complete loss of expression that can be regained by adding a single STAT 4n binding site from an unrelated gene, demonstrating STAT's direct involvement in *sna* regulation. It is interesting to note that the position where we inserted the new STAT site is on the opposite end to where the endogenous STAT sites are located, indicating there is flexibility on STAT protein localisation with respect to other transcriptional regulators binding to the enhancer, something also noticed for the *vvl1+2* STAT-regulated *cis*-regulatory element [20].

The activation of Upd in the gland primordium at st11 is very transient while *sna* transcription is maintained at least until st16 of embryogenesis, indicating that JAK/STAT signalling is only required for *sna*'s initial activation in the CA and PG primordia but not for its maintenance. Maintenance of the *sna-rg-R2P2* enhancer must be achieved by other elements of the gland gene-network induced by STAT or the Hox proteins. The existence of such a maintenance mechanism could explain why in wild type embryos, after mutating all three canonical STAT sites in the *sna-rg-R2P2 STATmut* reporter, the expression is still maintained in the CA: In these wild type embryos the endogenous gland gene-network is still functioning, activating the proposed ring gland maintenance mechanism that would be able to act on the *sna-rg-R2P2*

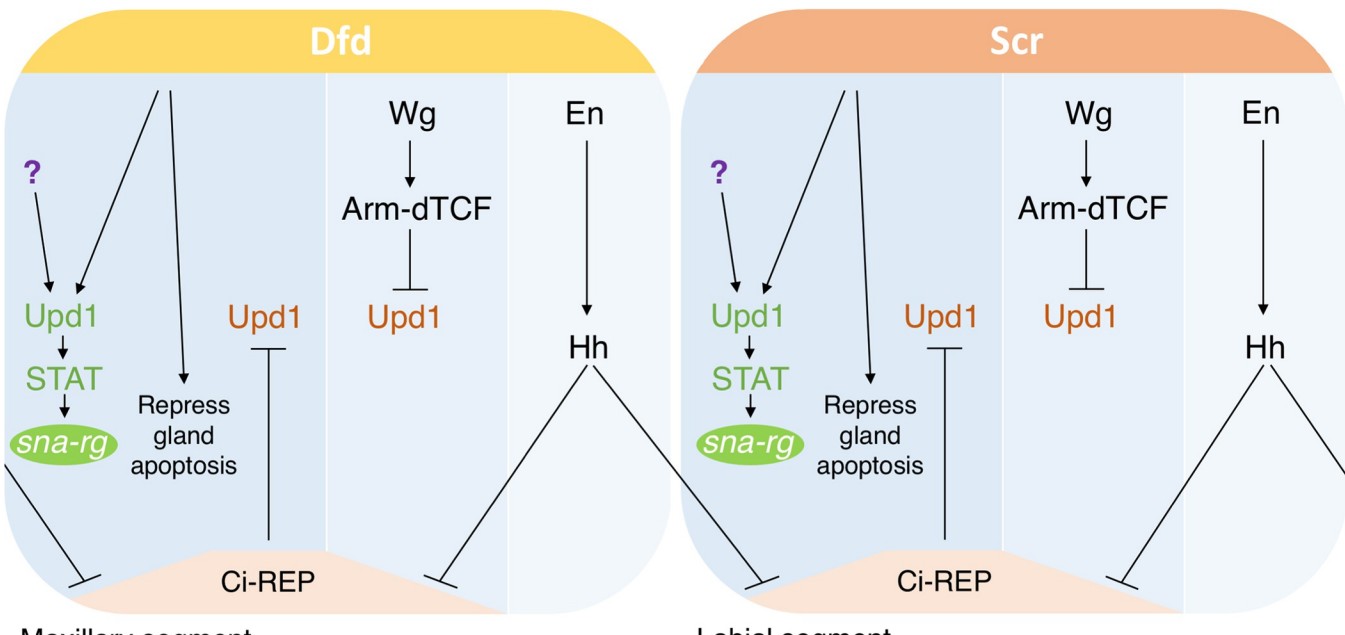

**Fig 7. Regulatory gene-network controlling CA and PG specification.** The regulation of the *sna-rg* enhancer in wild type embryos uncovers upstream factors mediating CA and PG specification. *upd* transcription is repressed directly or indirectly by both Ci and Arm-dTCF. Hh signalling from the posterior compartment releases the Ci repressive activity over *upd* transcription. Dfd and Scr proteins prevent the gland primordia from entering into apoptosis and also up-regulate Upd expression. Activated STAT induces *sna* transcription by direct binding to the *sna-rg* enhancer. Besides the Hox proteins, an additional factor (question mark) also contributes to *upd* expression in the maxilla and labium. This additional factor could be the activated form of the Ci protein or another factor still to be defined.

*STATmut* reporter even if it lacked the early STAT input. Although we cannot completely discard that the expression from the *sna-rg-R2P2 STATmut* reporter could be caused by the presence of cryptic STAT-binding sites not mutated in the *sna-rg-R2P2 STATmut-GFP* construct, two reasons favour the maintenance hypothesis. First, STAT activation in the gland primordia is very brief. Second, in embryos carrying both reporter constructs, the *sna-rg-R2P2* enhancer is activated earlier than the *sna-rg-R2P2 STATmut* enhancer (Fig 6I and 6J), indicating that these DNA binding site mutations can prevent the early STAT activation but not the later maintenance input. This CA maintenance input is most likely mediated by the 278bp A2 region as, in contrast to the *sna-rg-R2P2 STATmut* reporter, the *A1+STAT mut* reporter does not retain any CA expression. In *Df(1)os1A* embryos lacking all JAK/STAT ligands the endogenous gland gene-network is not activated and the *sna-rg-R2P2 STATmut* enhancer is completely silent as it lacks both the initiation and the maintenance inputs (Fig 6K). A similar gene-network feed-back loop acting on a STAT-regulated enhancer during organogenesis has already been reported [37].

## Regulatory similarities between CA, PG and tracheal specification

The CA, the PG and the tracheae have been proposed to originate by the divergence of an ancient serially repeated organ present in an arthropod ancestor [5]. If this was the case, specification of all three organs would be expected to be under similar upstream regulation. Using the early activation of *sna* transcription in the gland primordia through the *sna-rg* enhancer as a proxy for their specification, we found that the CA and PG primordia require the same signalling pathways controlling the specification of the tracheal primordia. JAK/STAT pathway activity is key for the activation of *vvl* and *trh* in the trachea and also for *sna* in the glands.

Moreover, direct STAT binding to a tracheal specific early enhancer is required for *vvl* activation and here we show the direct STAT binding is also required for *sna* specific expression in the glands. Similarly, the Wnt pathway that is required for restricting *vvl* and *trh* expression to the tracheal primordium is also restricting the spatial expression of *sna* in the cephalic segments, although the *sna-rg* ectopic activation observed in *wg* mutants is less pronounced than that observed for *vvl* and *trh* in the trachea, that in some instances results in a continuous tracheal pit stretching from T2 to A8 due to the fusion of the primordia in neighbouring segments [12,19,20].

Previous work has reported the requirement of Hh for *vvl* expression in the trachea and for tracheal branch specification [12,43]. Here we also find that Hh and En are required in the gland primordia, although we find this requirement to be more pronounced in the glands than in the trachea judging from the almost complete disappearance of *sna-rg* expression. We also found that in *hh* mutants, *upd* expression disappears in gland primordia but not in tracheal cells, supporting the idea that Hh requirement is stronger in glands than in trachea.

Ectopic trunk Hox protein expression can activate both *vvl* and *trh* tracheal expression in the head. Hox requirement for *vvl* expression is fundamental both in the glands and in the tracheal primordia being controlled via direct DNA binding sites [19]. Although anterior Antp, Ubx, Abd-A or Abd-B Hox expression can also induce ectopic *trh* activation, it is unclear if this is mediated through direct Hox binding to the DNA regulatory sites [5]. Similarly, the Dfd and Scr Hox proteins are required for gland formation and their ectopic expression can induce ectopic *sna-rg* activation in the tracheal cells [5]. Our results indicate that Hox requirement for *sna* activation may be indirect, as mutating the putative Hox binding sites in the enhancer does not affect its expression. The CA and the PG primordia co-express *sna-rg-GFP* and Dfd and Scr briefly during st11 at the very early specification stage, with Hox expression becoming undetectable in the glands when they initiate migration [5]. Our observation that the transient *upd* expression in the gland primordia is affected in *hh*, *wg* and *Dfd Scr* mutant embryos, suggests that these pathways regulate *sna* expression in the ring gland indirectly through the activation of JAK/STAT signalling.

Another interesting similarity between glands and trachea is that, although ectopic Hox gene expression can ectopically induce *sna-rg* and *trh* outside their normal domain, the lack of Hox expression does not completely abolish their endogenous expression, indicating that in both cases a second positive input can compensate for the absence of Hox mediated activation. Our results suggest that, at least in the glands, this redundant input could be provided by the activating Ci form (Figs 3G and 4I), but further analysis to confirm this possibility and discard alternative *sna-rg* activators should be performed.

## Differences between CA and PG specification

Development of the CA and the PG requires Sna activation, which in both primordia is regulated by the same enhancer. Also, both primordia are specified in the lateral ectoderm of the maxilla and the labium in cell clusters expressing *vvl* [5]. Despite these similarities, the position occupied by both primordia with respect to the *vvl* patch of expression is different. The CA is specified in the most ventral cells of the *vvl* maxillary patch and the PG is specified in the most dorsal cells of the *vvl* labial patch (Fig 5A). This suggests that despite their sharing of Hh, Wnt and JAK/STAT pathway regulation, the expression in each primordium must also have differential regulation. We have been unable to separate a CA enhancer from a PG enhancer by dissecting the *sna-R2P2 cis*-regulatory module into smaller fragments, suggesting that any gland specific binding sites in the enhancer are probably interspersed with the shared ones. The only case in which we were able to affect expression in the PG without affecting the CA was after

mutating the three STAT binding sites. Our results indicate *sna* expression in the CA is under direct STAT regulation (Fig 6C, 6G and 6H), as it is in the PG. The persistence of expression in the CA of the *sna-rg-R2P2 STATmut* reporter gene can be explained by the existence of differing expression maintenance factors in the CA and in the PG. Gland specific transcription factors like Seven-up (Svp) in the CA and Spalt (Sal) in the PG have been described that are expressed in the early gland primordia when they start their migration [5,44,45]. Future studies will help to discover if these or other gland specific factors are responsible for controlling the maintenance of *sna-rg* enhancer as well as the slightly different dorso-ventral positions where each gland is specified.

Our analysis of *snail* activation in the CA and PG shows that these glands and the trachea share similar upstream regulators, reinforcing the hypothesis that both diverged from an ancient segmentally repeated organ. In *Drosophila melanogaster* the CA and the PG primordia experiment a very active migration after which they fuse to the *corpora cardiaca* forming the ring gland [6]. This differs from more basal insects where the CA fuses to the *corpora cardiaca* but not to the PG, and from the Crustacea where the three equivalent glands are independent of each other [2–4,46]. As the mechanisms we here describe relate to the early specification of the glandular primordia in *Drosophila*, it will be interesting to investigate if the equivalent genes are also involved in the endocrine gland specification of more distant arthropods.

## Materials and methods

### Fly stocks

Flies were reared at 25°C on standard *Drosophila* medium. The following mutant alleles and transgenic lines were used: *wg^cx4*; CyO *wg^en11-lacZ*; *hh^AC*; *Df(2)en^E*; *ci^94*; *pan^2*; *Df(1)os1A*; *Dfd^16*; *Scr^4*; *Dfd^16 Scr^4 AbdB^M1*; *hth^P2*; *10xSTAT-GFP*; *arm-Gal4*; *sal-Gal4*; *UAS-arm^S10*; *UAS-upd*; *UAS-wg*; *UAS-ci^76*; *UAS-ci^PKA* [34] and *sna-GFP BAC* [26]. The following stocks previously generated in our laboratory were used: sna-*rg-GFP*; *sna-rg-mCherry*; *sna-rgR2P2-GFP*; *vvl1+2-mCherry*.

### Generation of *sna^ΔrgR2* deletion

The *snail-rg R2* enhancer was removed by CRISPR-Cas9 site-directed deletion. Two flanking *sna-rg* sgRNAs (S1 Table) where cloned into the directed insertion vector pCDF4 [47] and the constructs injected into the 25C (#B25709) or the 68A (#B25710) landing sites using the phiC31 standard method at the Drosophila Consolider-Ingenio 2007 Transformation platform, CBMSO/ Universidad Autónoma de Madrid. Germ line deletions were induced by combining the above *sna-rg* sgRNA transgenes with *nos*-Cas9 [47]. Putative heterozygous *mutant/If* males were individually crossed to *sna^1/CyO* females to identify mutations lethal over the null *sna^1* allele. Lethal alleles were tested by PCR to detect the generation of a deletion and the exact nature of the deletion was confirmed by sequencing using the FwdSeq snaCRISPR and RvsSeq snaCRISPR primers (S1 Table and S1F Fig).

### Immunostaining

Eggs laid overnight at 25°C or 29°C were dechorionated with 50% household bleach for 3 minutes and fixed with formaldehyde 4% and heptane in a 1:1 proportion for 20 minutes at room temperature. After removing the heptane, methanol was added and shaken vigorously to remove the vitelline membrane, embryos were stored for up to two weeks in PBS Tween 0,1%. The following primary antibodies were used: rabbit anti-GFP (Invitrogen, 1:300); rat anti-RFP (Chromotek, 1:200); mouse anti-β-Gal (Promega, 1:1000); mouse anti-En (Hybridoma Bank,

1:10); mouse anti-Scr (Hybridoma Bank, 1:25); rabbit anti-Dfd (Kauffman, 1:100); rabbit anti-Sal (our laboratory, 1:100); rabbit anti-cleaved Drosophila DCP1 (Cell Signalling, 1:100); mouse anti-Ci (Hybridoma Bank, 1:5); mouse anti-Wg (Hybridoma Bank, 1:25); sheep anti-DIG-AP (Roche, 1:2000). Secondary fluorescent antibodies from Invitrogen were used (1:200). Fluorescent images were obtained with Leica SPE confocal microscope and processed with ImageJ and Photoshop. Light microscopy samples were visualized in a Zeiss Axioplan microscope.

### In situ hybridization

155BS-*upd* from Doug Harrison and RE35237 from BDGP cDNAs were used to generate *upd* and *sna* RNA probes using the DIG RNA Labeling Kit (Roche). Secondary biotinylated antibody against mouse (1:200, Jackson ImmunoResearch) was used for double detection of RNA and protein.

### Constructs

*snaR2P2* subdivision avoided breaking sequence blocks conserved among different *Drosophila* species genomes (http://genome.ucsc.edu/). Fragments A1, A2 and A1+A2proximal were amplified by PCR using the following oligos: sna-rg for(KpnI) and snaR2P2B3rev; SnaR2P2B4for and sna-rg R2P2 rev (KpnI); or sna-rg for(KpnI) and snaR2P2B4rev respectively (S1 Table). Fragments were cloned into pGEMt-easy (Promega #A1360) and sequenced.

Fragment A1+A2distal was done using a modified protocol based on [48]. Template plasmid pGEMt-easy::SnaRGRP2 was amplified using primers SnaR2P2B3rev and SnaR2P2B5for flanking the deleted region. Before transformation, the PCR product was incubated 2h at 37˚C with DpnI restriction enzyme, polynucleotide kinase PNK, ligase and ligase buffer 2X.

pGEMt-easy::SnaRG-A1, pGEMt-easy::SnaRG-A2 and pGEMt-easy::SnaRG-A1+A2proximal were digested with Asp718 and subcloned into pCasper-PH-eGFP; pGEMt-easy::SnaRG-A1+A2distal were digested with NotI and subcloned into pCasper-PH-eGFP.

The pCASPER-PH-eGFP::A1+STAT-WT and pCASPER-PH-eGFP::A1+STAT-mut constructs were done using a pair of annealed oligos: statS1-NotI-for and statS1-NotI-rev; or statS1mut-NotI-for and statS1mut-NotI-rev respectively. Annealed oligos containing the STAT binding site and overhang NotI sites were cloned into pCASPER-PH-eGFP:: SnaRGR2P2A1 or pCASPER-PH-eGFP::SnaRGR2P2A2 NotI-digested.

The *sna-rg-R2P2 STATmut* enhancer was made by performing a PCR mutagenesis in two steps: a first PCR round with the *Fwd sna-rg + R2P1 Rvs OP 92E mut* primers and with the *Fwd SD 92E OP mut + Rvs sna-rg R2P2 92E mut* primers (S1 Table). The two resulting PCR amplicons containing overlapping sequences, were mixed and used as a template to be amplified with the external primers: *Fwd sna-rg BamHI + Rvs sna-rg R2P2 92E mut BamHI*, to generate the final *snail-R2P2 STATmut* enhancer fragment, which was subcloned into the BamHI site from a pCaSpeR modified version to simultaneously express *H2B-mRFP-P2A-LifeActinGFP*.

### Hox and STAT binding site mutagenesis

The putative class II (TGACAAAT) Hox binding site located at position 222–229 of *A1+A2proximal* was mutated to either erase the Hox site or to modify its affinity towards another Hox binding class site as described in [40]. Mutations were induced according to [48] using the following oligos: sna mut1 for and sna mut1 rev; sna mut2 for and sna mut2 rev; sna mut3 for and sna mut3 rev or sna mutNull for and sna mutNull rev.

The STAT binding site located at 504–513 (TTCCAATGAA) was mutated using the same method, with Sna mutSTAT for and Sna mutSTAT rev oligos.

## Supporting information

**S1 Fig. Isolation of a minimal *sna-rg* enhancer. (A)** Scheme of DNA elements tested indicating the location of the STAT binding sites. The *sna-rg R2* reporter comprises the *cis*-regulatory sequence deleted in the CRISPR-Cas9 *sna$^{\Delta rgR2}$* mutation. Expression at st11 of the full *sna-rg-GFP* **(B),** *sna-rg R1-GFP* **(C),** *sna-rg R2-GFP* **(D),** and *sna-rg R2P2-GFP* **(E)** constructs**. (F)** Scheme showing the fragment deleted in the *sna$^{\Delta rgR2}$* mutation indicating the sequence of the sgRNAs used and the sequences flanking the deletion.
(TIF)

**S2 Fig. Dynamic *upd* expression in wild type embryos.** Whole mount *in situ* RNA expression in 5-7h wild type embryos. Right panels show close ups of the maxillary and labial segments. **(A)** At st9 *upd* is expressed in stripes, located posterior to the En stripe **(B). (C)** At st10 there is a transient *upd* anteroposterior stripe running along the lateral ectoderm. **(D)** Close up of a st10 embryo focusing at the maxillary and labial segments. **(E-E')** At early st11 lateral ectoderm expression is detected in the gland primordia and in the tracheal pits. **(F-F')** At late st11 *upd* expression disappears from the CA and is detected in the PG primordium and in the invaginating trachea. Panels (B-D) show embryos double stained with anti-En (brown). Scale bars 50 μm.
(TIF)

**S3 Fig. Dfd, Scr, En and Wg expression in wild type and mutant embryos.** Dfd expression in **(A)** control heterozygous or in **(B)** *wg$^{CX4}$* and **(C)** *en$^E$* homozygous embryos. Scr expression in **(D)** control heterozygous or in **(E)** *wg$^{CX4}$* and **(F)** *en$^E$* homozygous embryos. En expression in **(G)** control heterozygous or **(H)** *Dfd$^{16}$ Scr$^4$* embryos. Wg expression in **(I)** control heterozygous or **(J)** *Dfd$^{16}$ Scr$^4$* embryos. Images show lateral views of the mandibular to the T1 segment in st11 embryos. Scale bars 50 μm.
(TIF)

**S4 Fig. Hox regulation of the *sna-rg* enhancer.** Expression of *sna-rg-R2P2* minimal enhancer in a *Dfd Scr* double mutant **(A)** or in a *hth$^{P2}$* mutant **(B). (C)** Hox binding site modifications introduced in the *sna-rg A1+A2proximal* constructs beside a heat map indicating their SELEX-seq DNA-binding affinity preferences. **(D)** Embryo expressing simultaneously a *sna-rg-mCherry* construct (D" red) and *sna-rg A1+A2proxHoxNull-GFP* (D' green) where the putative DNA binding site has been changed to a sequence not recognised by Hox-Exd-Hth *in vitro*. The expression of the *sna-rg A1+A2proxHoxNull-GFP* construct in the ring gland is not affected by the putative binding site mutation as well as maintaining the ectopic ventral expansion normal to a *sna-rg A1+A2prox* reporter. (E) *sna-rg A1+A2prox strong class3Hox-GFP*. (F) *sna-rg A1+A2prox strong class2Hox-GFP*. (G) *sna-rg A1+A2prox strong class1Hox-GFP*. Mutant embryos in (E-G) show strong expression in the maxilla and labium and the expression in other segments is barely affected. Panel C is modified from [40]. Embryos in (A) are also stained with anti-Scr to recognise the homozygous mutants. Scale bars 50 μm.
(TIF)

**S1 Table. Oligo sequences used in this study.**
(DOCX)

## Acknowledgments

We thank the Bloomington *Drosophila* Stock Center and the Sotillos, Casares, Couso and Kaufman's laboratories for sharing stocks and reagents.

## Author Contributions

**Conceptualization:** James C-G Hombría.

**Formal analysis:** Mar García-Ferrés, Carlos Sánchez-Higueras, Jose Manuel Espinosa-Vázquez, James C-G Hombría.

**Investigation:** Mar García-Ferrés, Carlos Sánchez-Higueras, Jose Manuel Espinosa-Vázquez.

**Project administration:** James C-G Hombría.

**Supervision:** James C-G Hombría.

**Writing – original draft:** Mar García-Ferrés, James C-G Hombría.

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
