## [Decision Letter · Decision Letter 0]

12 Sep 2022

Dear Dr Castelli-Gair Hombría,

We are pleased to inform you that your manuscript entitled "Specification of the endocrine primordia controlling insect moulting and metamorphosis by the JAK/STAT signalling pathway" has been editorially accepted for publication in PLOS Genetics. Congratulations!

The revised manuscript was seen by two of the original review commons referees; as you will see, both are enthusiastic about moving forward.

Yours sincerely,

Gregory S. Barsh

Editor-in-Chief

PLOS Genetics

Gregory Copenhaver

Editor-in-Chief

PLOS Genetics

Comments from the reviewers (if applicable):

Reviewer's Responses to Questions

**Comments to the Authors:**

Reviewer #1: The authors have addressed all of my comments and manuscript is now ready for publication.

Reviewer #2: The authors have answered most of my questions/concerns and I am satisfied with the revisions. I also find their explanations for the "prefer not to do" experiments acceptable as well. Manuscript is ready to go.

**Have all data underlying the figures and results presented in the manuscript been provided?**

Reviewer #1: Yes

Reviewer #2: None

PLOS authors have the option to publish the peer review history of their article (what does this mean?). If published, this will include your full peer review and any attached files.

Reviewer #1: **Yes: **Aleksandar Popadić

Reviewer #2: No

**Data Deposition**

http://datadryad.org/submit?journalID=pgenetics&manu=PGENETICS-D-22-00973

**Press Queries**

---

## [Editor Report · Acceptance letter]

26 Sep 2022

PGENETICS-D-22-00973 

Specification of the endocrine primordia controlling insect moulting and metamorphosis by the JAK/STAT signalling pathway 

Dear Dr Castelli-Gair Hombría, 

We are pleased to inform you that your manuscript entitled "Specification of the endocrine primordia controlling insect moulting and metamorphosis by the JAK/STAT signalling pathway" has been formally accepted for publication in PLOS Genetics! Your manuscript is now with our production department and you will be notified of the publication date in due course.

With kind regards,

Agnes Pap

PLOS Genetics

On behalf of:
